# The Modulatory Effects of Fatty Acids on Cancer Progression

**DOI:** 10.3390/biomedicines11020280

**Published:** 2023-01-19

**Authors:** Annemarie J. F. Westheim, Lara M. Stoffels, Ludwig J. Dubois, Jeroen van Bergenhenegouwen, Ardy van Helvoort, Ramon C. J. Langen, Ronit Shiri-Sverdlov, Jan Theys

**Affiliations:** 1The M-Lab, Department of Precision Medicine, GROW-School for Oncology and Reproduction, Maastricht University, 6229 ER Maastricht, The Netherlands; 2Department of Genetics and Cell Biology, NUTRIM-School of Nutrition and Translational Research in Metabolism, Maastricht University, 6229 ER Maastricht, The Netherlands; 3Advanced Health and Science Group, Danone Nutricia Research, 3584 CT Utrecht, The Netherlands; 4Division of Pharmacology, Faculty of Science, Utrecht Institute for Pharmaceutical Sciences, Utrecht University, 3584 CT Utrecht, The Netherlands; 5Department of Respiratory Medicine, NUTRIM School of Nutrition and Translational Research in Metabolism, Maastricht University, 6229 ER Maastricht, The Netherlands

**Keywords:** cancer, tumor progression, diet, fatty acids, inflammation

## Abstract

Cancer is the second leading cause of death worldwide and the global cancer burden rises rapidly. The risk factors for cancer development can often be attributed to lifestyle factors, of which an unhealthy diet is a major contributor. Dietary fat is an important macronutrient and therefore a crucial part of a well-balanced and healthy diet, but it is still unclear which specific fatty acids contribute to a healthy and well-balanced diet in the context of cancer risk and prognosis. In this review, we describe epidemiological evidence on the associations between the intake of different classes of fatty acids and the risk of developing cancer, and we provide preclinical evidence on how specific fatty acids can act on tumor cells, thereby modulating tumor progression and metastasis. Moreover, the pro- and anti-inflammatory effects of each of the different groups of fatty acids will be discussed specifically in the context of inflammation-induced cancer progression and we will highlight challenges as well as opportunities for successful application of fatty acid tailored nutritional interventions in the clinic.

## 1. Introduction

Cancer is the second leading cause of death globally, accounting for approximately 10 million deaths in 2020 [1], with the global cancer burden rising rapidly and predicted to reach 28.4 million cases in 2040 [2]. The risk factors for cancer development can often be attributed to lifestyle factors such as smoking, unhealthy diet, physical inactivity and alcohol consumption [3,4]. An unhealthy dietary composition and overeating are leading factors for the development of obesity and recent data indicate that obesity increases the risk for at least 13 different cancer types [5,6]. In addition to the risk of tumorigenesis, obesity is associated with a weak adverse effect on the survival of cancer patients [7]. In relation to modifiable lifestyle factors, there are multiple studies indicating that nutrition and diet have a large impact on cancer risk and progression [8,9,10]. The European Prospective Investigation into Cancer and Nutrition cohort (EPIC) is one of the largest prospective cohort studies in the world analyzing the relation between nutrition and cancer risk and progression [11], leading to numerous publications about the role of diet in cancer prevention and prognosis [12,13,14,15]. Recently, a systematic review of all published EPIC cohort studies (up to March 2021) addressing the effects of diet-related factors on the incidence and mortality of colorectal, breast, lung and prostate cancer has been published. Results indicated that fruit, vegetable, fatty fish, calcium and yogurt consumption had protective effects, while higher consumption of red and processed meat and soft drinks occurred as risk factors for cancer development or mortality [16]. These data suggest that a well-balanced and healthy diet can reduce the risk of cancer development and improve the prognosis of cancer patients.

Both macronutrient (proteins, carbohydrates, lipids and fibers) and micronutrient (vitamins and minerals) distribution are essential in a well-balanced and healthy diet. According to the World Health Organization (WHO) and the Food and Agriculture Organization of the United Nations (FAO), dietary fat should provide 15 to 35% of the energy for the healthy adult population, implying that dietary fatty acids are a crucial part of a well-balanced and healthy diet [17]. Fatty acids are often categorized into short-chain, medium-chain or (very) long-chain fatty acids. Short-chain fatty acids (SCFAs) are mainly generated upon intestinal fermentation of non-digestible dietary fibers [18], whereas medium- and (very) long-chain fatty acids are mostly taken directly from the diet [17]. Medium- and (very) long-chain fatty acids can be classified into saturated (SFAs), monounsaturated (MUFAs) or polyunsaturated fatty acids (PUFAs) and the latter are further sub-classified into omega-3, 6 or 9 (*n*-3, *n*-6 or *n*-9) MUFAs/PUFAs. Additionally, MUFAs and PUFAs also appear as trans-fatty acid (TFAs), which are classified into industrial and ruminant TFAs (Figure 1).

At the cellular level, fatty acids are acquired from a range of intracellular and extracellular sources. When fatty acids are taken up by cells in the body, they are activated by esterification with co-enzyme A to form fatty acyl-co enzyme A (FA-CoA). Following this, FA-CoA can be desaturated or elongated, or used for lipid synthesis, energy production and protein acylation [19]. Accordingly, dietary fatty acids and their metabolites fulfil a crucial role in regulating energy metabolism, structural cell integrity, cellular signaling and immune functioning [20,21,22]. Not surprisingly, disruptions in fatty acid metabolism are involved in several hallmarks of cancer, including proliferation, migration, angiogenesis, anti-tumor immunity, and tumor-promoting inflammation [19,23,24,25]. Tumors are highly heterogeneous, as apart from cancer cells, the tumor is highly infiltrated by immune cells, including tumor-associated macrophages, lymphocytes, myeloid-derived suppressor cells, neutrophils and dendritic cells, often creating an immunosuppressive tumor micro-environment (TME) [26]. Cancer-associated fibroblasts support a pro-tumorigenic TME by matrix deposition and production of soluble factors that recruit immune cells and promote angiogenesis [27]. The disorganized tumor vascularization also contributes to a pro-tumorigenic TME characterized by hypoxia and acidosis due to nutrient depletion [28]. The TME can be influenced by various components, such as hypoxia-activated prodrugs, biomaterials and dietary interventions [29,30,31]. In recent years, an increasing number of publications have explored the relevance of lipids to tumor growth and their effects on TME [32,33,34]. The use of specific pharmacological inhibitors designed to target enzymes in *de novo* fatty acid synthesis and exogenous lipid uptake in cancer cells in order to inhibit tumor progression has already been extensively reviewed elsewhere [35]. Another potentially effective approach to reducing tumor progression involves the use of fatty acid tailored dietary interventions, as specific fatty acids interfere with signal transduction and metabolic pathways in tumor and immune cells [36].

This review aims to evaluate the interplay between the dietary intake of specific fatty acid groups and cancer initiation, progression (main focus), and metastasis (Figure 2). We will report epidemiological evidence on the associations between the intake of different classes of fatty acids and the risk of developing cancer, and we will discuss preclinical evidence on how specific fatty acids can influence tumor cells in a direct manner to modulate tumor progression and metastasis. Moreover, we will focus on the indirect effects of each of the different groups of fatty acids on tumor progression by describing the impact of the fatty acids on immune cell functioning, anti-tumor immunity and tumor-promoting inflammation (Figure 2; Table 1). Finally, we will highlight challenges as well as opportunities for the successful application of fatty acid-tailored nutritional interventions in the clinic.

## 2. SFAs Stimulate Tumor Progression

SFAs occur naturally in many food products, including (red) meat, dairy products and coconut and palm oil. The tumor-promoting effects of SFAs have been detected in numerous large-scale human cohort studies reporting positive associations between SFA intake and breast, prostate and overall cancer risk [104,105,106,107]. Surprisingly, several meta-analyses did not find associations between high SFA intake and the risk of developing colon, epithelial ovarian or endometrial cancer [108,109,110,111]. One possible reason for this discrepancy is that the association between SFA intake and cancer risk could be dependent on the type of cancer studied. Moreover, the food source rich in SFAs could possibly be responsible for the observed differences, with SFAs from (red) meat being probably pro-carcinogenic [112,113]. However, most human cohort studies do not distinguish between the intake of different SFAs or their sources. These data underline the need to study each different SFA and its derived food source in cancer risk and progression in more detail.

Preclinical data have indicated that a high dietary intake of myristic and palmitic acid stimulates tumor growth in mice [37,38]. Cells use SFAs for post-translational protein acylation to regulate multiple signaling pathways that promote cellular growth and survival [114,115,116,117] and in tumor cells, this process can contribute to tumor growth [37,38]. For example, in a subcutaneous mouse model of prostate cancer, a high dietary intake of myristic acid promoted tumor growth via myristoylation of Src kinases [37]. In a comparable prostate cancer mouse study, high palmitic acid intake led to elevated levels of exogenous palmitoyl-CoA, enhancing Src kinase-mediated oncogenic signaling in tumor cells [38]. In line with these data, approaches to inhibit post-translational protein palmitoylation and myristoylation have already shown therapeutic potential by inhibiting the tumor-initiating capacity of cancer stem cells and blocking prostate cancer progression [37,118,119].

In addition to the use of SFAs for post-translational protein acylation, cells also use SFAs as building blocks for membrane lipids. The plasma membrane forms a selective barrier controlling the transport of molecules inside and outside the cell, regulates cell communication and is involved in proliferation, differentiation, secretion, migration, invasion and phagocytosis. Of note, the composition of the plasma membrane is essential in these processes and SFAs, as one of the major fatty acid classes, hold a prominent role in a balanced membrane fatty acid profile [120,121]. Dietary intake influences the amount of SFAs in lipid membranes: in healthy rat tissue, analysis of the relationship between dietary fat intake and tissue fatty acid composition indicated that SFA content, despite large changes in diet, remained constant in heart, brain, liver and muscle tissue, while adipose tissue was highly responsive to dietary fatty acid intake [122]. Currently, it is not yet known whether tumor tissue is responsive to changes in the dietary intake of specific SFAs and/or alteration in total SFA dietary content. In addition to dietary intake, the expression of desaturation enzymes also influences the amount of SFAs in membrane lipids and in several cancer types, downregulation of stearoyl-CoA desaturase (SCD) 5 has been observed [123,124,125], while SCD1 is often overexpressed [123,126,127]. This means that the conversion rate of SFAs into MUFAs may be altered in cancerous tissue in comparison to healthy tissue, which may influence the amount of SFAs present in the membranes of tumor cells. Specific SFAs have been reported to be increased in the plasma membrane of tumor cells [123,128] and this may result in decreased membrane fluidity, which may hinder passive diffusion of anti-cancer therapeutics, eventually leading to multi-drug resistance and cancer progression [129]. However, to maintain proliferation, cancer cells also need to convert SFAs into MUFAs to reduce the intrinsic cytotoxicity of SFA and inhibition of SCD enzymes has been shown to worsen palmitic acid cytotoxicity [21,130]. Moreover, decreased SFA membrane content also stimulates metastatic properties due to increased membrane fluidity [131] (Figure 2). Apart from the membrane fluidity, metastatic properties of tumor cells are also dictated by epigenetics and data suggest that high levels of dietary metabolites of palmitic acid can induce stable transcriptional and chromatin alterations, leading to a pro-metastatic phenotype in cancerous cells [40].

Tumor-promoting inflammation is one of the hallmarks of cancer and palmitic acid is known to induce pro-inflammatory NLR family pyrin domain containing 3 (NLRP3) inflammasome activation, nuclear factor-κB (NF-κB) and toll like receptor 4 (TLR4) dependent signaling [39,41,42,45], which can mediate pro-carcinogenic effects. In vivo, in a mouse model of liver cancer, a diet rich in SFAs (mainly metabolized into palmitic acid) increased the incidence of liver cancer and hepatocyte proliferation, which was accompanied by increased liver macrophage infiltration as well as by elevated NF-κB, cyclin D1, tumor necrosis factor (TNF) and interleukin (IL)-1 expression levels in the livers [39]. Additional recent evidence suggests that palmitic acid-mediated TLR4-dependent pro-inflammatory signaling is specifically relevant in the context of cancer progression. TLR4, IL-6 and TNF-α expression was upregulated by palmitic acid in a neuroblastoma cell line in vitro and in vivo, TLR4^-/-^ colorectal cancer (CRC) tumor growth was not affected by a diet enriched with palmitic acid, while wild-type CRC tumor growth was increased upon dietary intervention in mice [43,44]. Together, these data point to the potential detrimental pro-inflammatory effects of SFAs, specifically palmitic acid, in promoting tumor progression (Figure 2).

## 3. Effects of MUFAs on Tumor Progression Are Inconclusive

A traditional Mediterranean dietary pattern is characterized by a high consumption of MUFAs, mainly derived from olive oil. Epidemiological data on the association between MUFA intake and the risk for cancer indicate that MUFAs originating from plants, such as olive oil, have been associated with reduced breast cancer risk and significantly lower cancer mortality, while a high intake of animal-derived MUFAs has been associated with an elevated cancer risk [132,133,134,135,136]. The impact of MUFAs on tumor progression may also be caused by differences in the ratios of absolute total fat/MUFA intake vs. the relative intake of total fat/MUFAs (e.g., when MUFAs are consumed via meat, these ratios change due to the SFA content in meat). Therefore, to understand the potential impact of MUFAs on cancer risk and progression, it will be important in future studies not only to study the role of the various types of different MUFAs but also their food source (plant- or animal-derived MUFAs) and the ratios of absolute and relative total fat/MUFA intake.

In the context of tumor progression, both pro- and anti-carcinogenic effects have been reported. Pro-carcinogenic effects of oleic acid-based diets in mice include promotion of tumor cell growth and migration in cervical cancer and increased tumor burden and angiogenesis in pancreatic cancer [47,48]. In contrast, in several lung cancer mouse models, oleic acid-containing diets have shown beneficial effects, such as reduced tumor incidence, suppressed cancer cell proliferation, increased tumor latency and survival and inhibition of formation of metastatic lesions in the liver [49,50,51]. Apart from the different animal models and read-out parameters used in these studies, another possible explanation for these contradictory data may be the contributing food source of oleic acid [135]. Currently, in the majority of animal studies, the MUFA source is not described; hence, it will be essential for future studies to report this information to gain more insights into the potential causal effects and mechanisms of MUFA intake on tumor progression.

One of the mechanisms through which MUFAs potentially modulate tumor progression is related to epigenetic alterations. Dietary MUFAs might disrupt epigenetic patterns in cancer cells, including altered DNA methylation profiles and aberrant histone modifications. For example, it has been suggested that dietary administration of olive oil, rich in oleic acid, may induce higher levels of global DNA methylation and histone modifications in breast cancer cells compared to a diet based on corn oil [137]. Another mechanism through which MUFAs might modulate tumor progression is related to MUFAs being used as building blocks for membrane lipids: upon incorporation into membrane lipids, MUFAs influence the physicochemical properties of the membrane, which may alter several of its functions, e.g., alterations in the regulation of cell communication, proliferation, secretion and invasion [138]. Dietary intake influences the membrane MUFA content, with heart, liver and muscle tissues being only slightly responsive to dietary MUFA content, while the MUFA content in the brain, potentially due to the role of oleic acid in myelination in the brain [139], is strictly regulated in response to dietary MUFA content [122]. Currently, it is not yet known to what extent the membranes of tumor cells are responsive to dietary MUFA content changes. The MUFA membrane content is also partly regulated by SCD enzymes and as indicated before, SCD enzymes are differentially expressed in cancerous tissue [123,124,125,126,127], due to which the conversion rate of SFAs into MUFAs may be altered, leading to changes in the amount of MUFAs present in the membranes of tumor cells. Several studies have described specific MUFAs to be increased in the plasma membrane of human tumor cells [123,128], and increased MUFA content in the plasma membrane of tumor cells not only potentiates metastatic properties by increasing membrane fluidity [131] (Figure 2), but an increased conversion of SFAs into MUFAs also prevents cancer cells from the intrinsic cytotoxicity of SFAs, thereby stimulating cancer cell survival [140].

MUFAs may also indirectly influence tumor progression by modulating tumor-promoting inflammation. Oleic acid is able to counteract the pro-inflammatory effects of SFA stearic acid in human aortic endothelial cells in vitro and specifically in the context of cancer. Extracts from the avocado (*P. americana*) fruit, rich in oleic acid (75% of total fatty acids), have shown anti-inflammatory and anti-cancer activity against colon and liver cancer cells in a dose-dependent manner [52]. In addition to oleic acid, palmitoleic acid has also been reported to exert anti-inflammatory effects. Palmitoleic acid rescued the pro-inflammatory effects of lipopolysaccharide (LPS)-stimulated primary macrophages and reduced the expression of hypoxia-inducible factor-1α (HIF-1α), an important driver in several aspects of cancer [46]. In vivo in diabetic rats, dietary administration of fish oil low in *n*-3 PUFAs but highly enriched in oleic acid suppressed the production of the pro-inflammatory markers TNF-a and NF-kB [53,54]. Nonetheless, given the heterogeneous fatty acid composition of this dietary intervention, the observed anti-inflammatory effects could also be induced by other fatty acids [54]. More convincing are the clinical data indicating that the consumption of a MUFA-enriched intervention diet led to an anti-inflammatory gene expression profile in abdominally overweight subjects [55]. Overall, accumulating evidence links MUFAs to anti-inflammatory properties, which potentially contribute to a reduction in tumor progression (Figure 2).

## 4. A Diet with a High *n*-3/*n*-6 PUFA Ratio Reduces Tumor Progression

Two types of dietary PUFAs exist, the *n*-3 and *n*-6 PUFAs. The most common dietary sources of *n*-3 PUFAs are fish oil, vegetable oil, nuts and flaxseeds, while dietary intake of sources rich in *n*-6 PUFAs includes meat, poultry, eggs, sunflower oil and soybean oil. Epidemiological evidence indicates that the anti-inflammatory effects of the *n*-3 PUFAs eicosapentaenoic acid (EPA) and docosahexaenoic acid (DHA) decrease the risk of developing cancer [57,141,142,143] (Figure 2). A high intake of *n*-6 PUFA-rich diets is generally considered unhealthy, predominantly due to the pro-inflammatory effects of arachidonic acid (ARA) [70,71]. Nowadays, increasing evidence is arising showing that, rather than the absolute intake, it is the *n*-3/*n*-6 PUFA ratio that is of clinical relevance, as evidenced by the high intake ratios of *n*-3/*n*-6 PUFAs associated with a lower risk of cancer development [144,145].

In vivo studies have indicated that a diet enriched with DHA or fish oil reduces tumor growth in murine models of colon and prostate cancer [56,62]. Mechanistically, these studies indicated that a diet enriched with DHA increased the levels of *n*-3 PUFA peroxidation and that a diet enriched with fish oil led to elevated tumor markers for oxidative stress, resulting in reduced tumor growth [56,62]. In line with these in vivo data, HCT116 colon cancer cells in an acidic TME accumulate *n*-3 PUFAs in lipid droplets in vitro, increasing the extent of lipid peroxidation and eventually promoting ferroptosis [62] (Figure 2), a type of regulated cell death driven by iron-dependent phospholipid peroxidation [146]. These data underscore the relevance of *n*-3 PUFAs that are metabolized intracellularly, causing oxidative stress that contributes to the anti-carcinogenic effects of *n*-3 PUFAs. *n*-6 PUFAs were also shown to accumulate in lipid droplets in an acidic TME in colon cancer cells in vitro, also promoting ferroptosis by increasing the extent of lipid peroxidation [62]. Moreover, the anti-cancer activity of *n*-6 PUFAs has been reported in other in vitro studies, primarily through the induction of apoptosis as a consequence of excessive reactive oxygen species (ROS) production by activation of mitochondrial-mediated pathways [66,67,68] (Figure 2). However, such effects have not been confirmed in vivo; in contrast, *n*-6 PUFAs have tumor-promoting effects in some in vivo models of mammary and pancreatic cancer [64,65] and cause enhanced liver metastasis in pancreatic cancer tumor-bearing mice [64]. Possibly, antioxidants in vivo reduce ROS production, preventing the apoptosis of cancer cells induced by *n*-6 PUFAs, while the metabolites of *n*-6 PUFAs still induce tumor-promoting effects.

In addition, *n*-3 PUFAs have also been implicated in the epigenetic regulation of carcinogenesis. DHA has been shown in vitro to increase global H3 histone acetylation levels and reduce methylation of CpG sites in the DNA of several pro-apoptotic genes [63]. Dietary administration of fish oil in vivo protected the rat colon from carcinogen-induced oncogene expression by preventing miRNA dysregulation [147,148] as an additional epigenetic mechanism [149]. In contrast, chronic exposure to an *n*-6 PUFA-based high-fat diet (HFD) has been shown to induce several epigenetic changes that increase the risk of inflammation and colon carcinogenesis [69].

PUFAs also serve as building blocks for membrane lipids [150]. The PUFA content is altered in cancerous tissue [123,151], but there is no consensus yet on which specific PUFAs are differentially incorporated in malignant tissue in comparison to healthy tissue, although the majority of the studies reported an increase of *n*-6 PUFAs in malignant tissue membranes [123,151,152,153]. There is evidence that the PUFA lipid content can be influenced by nutritional intake [154,155], e.g., administration of a linoleic acid (*n*-6 PUFA) rich diet to prostate tumor bearing mice induced a 2.7-fold increase in the *n*-6/*n*-3 PUFA ratio in tumors, while feeding of a stearidonic acid (*n*-3 PUFA) rich diet led to a 4.2-fold decrease in this ratio (119). Upon a high dietary intake of EPA and DHA, these *n*-3 PUFAs are incorporated into membrane phospholipids, partially replacing ARA (*n*-6 PUFA). Although it is not known how this replacement influences membrane fluidity (Figure 2), the interaction between growth factors, cytokines and hormones with their receptors may be affected, resulting in changes in signal transduction and cell survival [154,156,157].

In addition to the use of PUFAs as building blocks for membrane lipids, PUFAs, *n*-3 as well as *n*-6, also serve as substrates for cyclooxygenase and lipoxygenase isozymes and are precursors for different eicosanoids, such as prostaglandins, leukotrienes and thromboxanes. Generally, eicosanoids derived from *n*-3 PUFAs have low or even anti-inflammatory properties, whereas eicosanoids derived from *n*-6 PUFAs induce pro-inflammatory effects (Figure 2). The anti-inflammatory eicosanoids derived from EPA and DHA not only decrease the risk of cancer development [57,141,142,143] but also slow down tumor progression. For instance, in a mouse model for breast cancer, HFD in combination with fish oil resulted in decreased tumor weight and number, accompanied by a decrease in the pro-inflammatory markers TNF-α and IL-6 and an increase in the anti-inflammatory marker IL-10 [58]. Furthermore, dietary supplementation with *n*-3 PUFAs has already been shown in several clinical trials to reduce chronic low-grade inflammation in cancer patients [59]. Dietary administration of EPA plus DHA reduced oxidative stress, decreased production of the pro-inflammatory markers C-reactive protein and IL-6 and also lowered serum levels of the pro-inflammatory mediator prostaglandin E2 (PGE2) in cancer patients [60,61]. In contrast, diets rich in *n*-6 PUFAs promote pro-inflammatory immune responses, thereby favoring tumor progression (Figure 2). For instance, long-term feeding of an *n*-6 PUFA-rich HFD in mice induced the accumulation of inflammatory and hyperplastic lesions [158], which predisposes to colorectal cancer [159,160]. Additionally, the *n*-6 PUFA-rich HFD diet also increased colonic cyclo-oxygenase-2 (COX2) expression, the enzyme responsible for the conversion of ARA into pro-inflammatory prostaglandin E2 (PGE-2). PGE-2 was shown to stimulate myeloid-derived suppressor cell (MDSC) accumulation, leading to inhibition of immuno-surveillance in the TME and eventually promoting tumor progression in a mouse model of mammary carcinoma [72].

Overall, PUFAs serve as building blocks for membrane lipids or can be metabolized intracellularly, and preclinical in vivo data indicate a beneficial effect of *n*-3 PUFAs via increased apoptosis and ferroptosis of tumor cells (Figure 2), while *n*-6 PUFA-rich diets have tumor-promoting effects. PUFAs also serve as substrates for cyclooxygenase and lipoxygenase isozymes for the biosynthesis of eicosanoids and *n*-3 and *n*-6 PUFAs generally compete for these isozymes, with *n*-3 PUFA-derived eicosanoids exhibiting anti-inflammatory effects reducing cancer progression, whereas *n*-6 PUFA-derived eicosanoids exhibit pro-inflammatory effects favoring a pro-tumorigenic environment (Figure 2). Altogether, these data indicate the relevance of *n*-3/*n*-6 PUFA ratio rather than their absolute levels, suggesting that a high ratio of *n*-3/*n*-6 PUFAs could reduce cancer progression by increasing apoptosis/ferroptosis of tumor cells and decreasing inflammation, one of the hallmarks of cancer.

## 5. Effects of TFAs on Tumor Progression Is Inconclusive

TFAs are defined as MUFAs or PUFAs with one or more double bonds in trans-configuration. There are 2 types of TFAs, the industrial and ruminant TFAs. Industrial TFAs (iTFAs) are made by the partial hydrogenation of vegetable or fish oils, with elaidic acid being the most common iTFA present in margarine and fried foods (up to 2% of total grams of fat). Ruminant TFAs (rTFAs) are produced by bacterial metabolism of PUFAs in the rumen from ruminant animals and therefore present in ruminant fat sources such as in cow and goat milk, beef and lamb (typically between 2–9% of total fatty acids), with conjugated linoleic acid (CLA) isomers, vaccenic acid and trans-palmitoleic acid being the most common rTFAs. Human studies analyzing the association between iTFA and/or rTFA intake and cancer risk showed mixed results depending on the specific TFA and cancer type [161,162,163] (Figure 2). Meta-analyses and systematic reviews reported no associations between total TFA and cis-9,trans-12-CLA intake and breast cancer risk [161,162], while a novel, large-scale, cohort study did find positive associations for CLA (isomer not indicated), palmitelaidic acid and elaidic acid intake and breast cancer risk, but observed no association for vaccenic acid intake [163]. Additionally, for prostate cancer, positive associations were reported for elaidic acid and trans-palmitoleic acid intake, but not for vaccinic acid and trans-10,cis-12 CLA [161]. Thus, the impact of different TFAs (i.e., industrial or ruminant) on cancer risk in general, as well as for specific cancer types, warrants further investigation.

Already, in the 1980s, it was recognized that dietary administration of elaidic acid to Ehrlich ascites tumor-bearing mice resulted in decreased survival [73,74]. More recently, elaidic acid has been described to enhance colorectal cancer cell growth and survival in vitro and dietary administration of elaidic acid in vivo has been shown to promote colon carcinogenesis [75,164]. Moreover, it was found that elaidic acid stimulated colon cancer metastasis in several mouse models by increasing cancer cell stemness and epithelial–mesenchymal transition [75,76,77] (Figure 2). In contrast to these iTFAs, multiple preclinical studies describe the therapeutic effects of specific rTFAs, specifically CLA isomers (i.e., cis-9,trans-11-CLA or trans-10,cis-12-CLA), via increasing apoptosis, suppressing proliferation and reducing metastasis [79,83,84,86], shown to be regulated via decreasing peroxisome proliferator-activated receptor (PPAR) γ and epidermal growth factor receptor (EGFR) signaling and via the suppression of matrix metalloproteinase (MMP) activity [79,84,85]. In contrast, dietary CLA administration has also been shown to lead to tumor progression and cellular expansion mediated through increased phosphorylation of the insulin-like growth factor 1 (IGF-I)/insulin receptor, as well as through increased signaling of the mitogen-activated protein kinase/ERK kinase (MEK)/extracellular-signal-regulated kinase (ERK) (MEK-ERK) and phosphoinositide 3-kinases–protein kinase B (PI3K-AKT) pathways in tumor cells [80,81,82]. Moreover, CLA supplementation has been shown to have a stimulatory effect on metastasis formation in mice [81]. Altogether, since preclinical data on the effects of rTFAs on cancer cells remain ambiguous (Figure 2), improved standardization of intervention designs and the use of appropriate experimental models is essential to allow systematic evaluation of the effects of these TFAs on cancer progression.

While preclinical data on the effects of rTFAs on cancer cells remain contradictory, there is evidence that rTFAs might execute therapeutic anti-cancer effects by means of suppressing inflammation (Figure 2). Non-toxic doses of vaccinic and trans-palmitoleic acid were reported to attenuate TNF and IL-8 gene expression in liver cancer cells, pretreated with TNF-α to mimic the effect of low-grade inflammation and non-toxic doses of CLA mixtures decreased TNF-α and IL-1β secretion and PGE-2 release in lung cancer cells [78,165], probably mediated through PPARα [165]. Moreover, CLA isomers also target immune cells with PPAR receptors, such as macrophages, which, upon CLA isomer treatment, decreased the production and activity of multiple mediators of inflammation [166,167]. Preclinical evidence regarding the anti-inflammatory effects of CLA isomers in in vivo cancer models is sparse. However, dietary CLA supplementation significantly decreased the risk of inflammation-induced colorectal cancer in mice through a PPARγ-dependent mechanism and daily oral administration of milk fat CLA led to a significant reduction in tumor weight and volume, accompanied by a significant restoration of the white blood cell count to near normal values, in a mouse model of Ehrlich ascites carcinoma [86,87]. Overall, these data point toward the therapeutic potential of CLA isomers in cancers associated with chronic low-grade inflammation, potentially via reducing systemic inflammation; however, this notion awaits further confirmation in additional preclinical models.

## 6. SCFAs Reduce Tumor Progression

Non-digestible dietary fibers are present in whole grain products, fruits, vegetables, beans, nuts and seeds, and epidemiological data indicate that high dietary fiber intake is inversely associated with the risk of developing cancer [168,169]. Non-digestible dietary fibers are fermented in the intestine by gut microbes into SCFAs such as acetate, propionate and butyrate [18]. Only limited cohort studies, with a small number of participants, have been performed to assess the correlation between fecal SCFA levels and cancer risk in humans and no significant associations have been reported [170,171]. Strikingly, associations between serum SCFA levels and cancer risk were demonstrated, with negative correlations for valeric acid and positive correlations for acetic and propionic acid [172]. These observed differences in the associations between fecal and serum SCFA levels and cancer risk can potentially be caused by the complex interplay between the production and absorption of SCFAs in the gastrointestinal tract.

In vivo studies have shown that butyrate, derived from a high fructo-oligosaccharide/inulin diet or a fiber-rich diet, reduced tumor development in a chemically induced colon cancer mouse model [92] and in a lymphoma mouse model [93]. In normal physiology, the majority of SCFAs derived from dietary fibers are rapidly absorbed in the gut and are used for energy production via SCFA oxidation. However, due to the Warburg effect, cancerous colonocytes have been shown to rely on glucose as their primary energy source. As a consequence, specifically within these cancerous colonocytes, which are highly dependent on the Warburg effect, SCFAs accumulate in the nucleus, where they exert anti-carcinogenic effects, instead of being used as an energy source [173]. In the nucleus, butyrate and, to a lesser extent, propionate and acetate exert anti-carcinogenic effects via histone deacetylase (HDAC) inhibition, resulting in epigenetic changes and modified gene expression [88], leading to suppressed cancer cell proliferation and increased apoptosis [94,95,96,97] (Figure 2). Since the induction of apoptosis was not observed in non-carcinogenic cells [174], this effect may be selective for cancer cells. In addition, butyrate suppressed the motility of colorectal cancer cells in an HDAC-dependent manner, revealing its potential anti-metastatic properties [98] (Figure 2). These HDAC-inhibiting anti-cancer effects of butyrate have also been confirmed in vivo in the chemically induced colon cancer mouse model [92] and in the lymphoma mouse model [93]. Overall, these preclinical data indicate that a diet rich in fibers, which are fermented into SCFAs, has the potential to reduce cancer progression via inhibiting HDAC activity.

In addition to reducing tumor progression via inhibiting HDAC activity, SCFAs may also indirectly reduce tumor progression, since SCFAs have been shown to exert anti-inflammatory properties, and tumor-promoting inflammation is one of the hallmarks of cancer [25,175,176]. Preclinical evidence demonstrated that SCFAs reduce inflammation-induced tumor growth in colon cancer, as well as in other cancer types [91,103,177] (Figure 2). More specifically, a deficiency of the butyrate receptor G-protein-coupled receptor (GPR) 109a promoted inflammation and colon carcinogenesis in mice and in line with these data, dietary inulin-type fructans led to reduced tumor cell infiltration in the liver and decreased inflammation-induced cancer progression in a leukemia mouse model, mediated by propionate [91,103]. Although the exact mechanism is not fully deciphered, it has been proposed that in the absence of SCFAs, inflammatory immune cells accumulate in the gut and local lymph nodes, from which they can either migrate to distant organs or secrete pro-inflammatory cytokines that can contribute to enhanced tumor growth in target organs [177]. SCFAs may also prevent or reduce (colon) carcinogenesis via the improvement of gastro-intestinal barrier function [178] (Figure 2). The gastrointestinal mucosa is essential in preventing epithelial translocation of pathogenic organisms to systemic organs, while maintaining adequate permeability for nutrient absorption, and disruption can cause microorganisms, toxins and food antigens to translocate into the bloodstream, leading to systemic inflammation [179,180,181]. In vitro studies have indicated that SCFAs significantly improve intestinal barrier function, as measured by transepithelial electrical resistance (TEER) [99,100,101] and one can speculate that this can potentially reduce inflammation-induced tumor progression, specifically in the context of colon cancer. Furthermore, SCFAs stimulate anti-inflammatory immune responses by triggering anti-inflammatory cell function, while suppressing pro-inflammatory immune cells. For instance, in mouse models of (ulcerative) colitis, SCFAs in drinking water had a protective effect by increasing the frequency and function of colonic regulator T-cells and reducing TNF-α as well as IL-17 production by peripheral mononuclear cells [89,90]. Oral butyrate administration also improved intestinal trophism, inhibited eosinophil and neutrophil infiltration in the colon and reduced the percentages of activated B cells, macrophages and T cells toward control levels in cecal lymph nodes and Peyer’ patches [102]. Although these data have been reported in mouse models of (ulcerative) colitis, they are also applicable in the context of colon cancer, as ulcerative colitis is a risk factor for colon cancer [182]. Overall, these preclinical data indicate that diets enriched with indigestible fibers, which are fermented into SCFAs, can reduce systemic inflammation and potentially contribute to decreased inflammation-induced cancer progression (Figure 2).

## 7. Concluding Remarks

In conclusion, as SCFAs and *n*-3 PUFAs reduce tumor progression by acting on tumor cells and suppressing chronic low-grade systemic inflammation, cancer patients could potentially benefit from dietary regimens rich in SCFAs or nutritional interventions with a high *n*-3/*n*-6 PUFA ratio (Table 1). The effects of MUFAs and rTFAs are not yet completely understood, while dietary regimens rich in SFAs and iTFAs should be avoided, as these stimulate cancer progression (Table 1). However, several challenges need to be overcome before dietary regimens focusing on SCFAs and high *n*-3/*n*-6 PUFA ratios can be integrated into the standard of care. The most favorable intake levels and ratios between the different SCFAs and *n*-3/*n*-6 PUFAs, as well as the best food source and different doses of these fatty acids, should be explored. Additionally, personalized diets need to be designed, since a single recommendation diet for all cancer patients will not be sufficient due to inter-individual variability in nutritional status, lipid metabolism, immune responsiveness and gut microbiome composition. Additionally, depending on the tumor type, disease stage, presence of comorbidities and the recommended treatment (e.g., chemotherapy, radiotherapy or immunotherapy), differential nutritional interventions will be advisable [183,184,185,186]. Furthermore, fatty acid mimetics bear huge potential for the treatment of cancer and await further validation in (pre)clinical studies [187]. Overall, despite promising preclinical in vitro and in vivo data supporting the use of SCFAs and *n*-3 PUFA/*n*-6 PUFA focused nutritional interventions as an approach to control tumor progression, several challenges remain to be overcome, highlighting the necessity of more studies as a basis to specifically advise on SCFA- or *n*-3 PUFA-tailored diets.

## Figures and Tables

**Figure 1 biomedicines-11-00280-f001:**
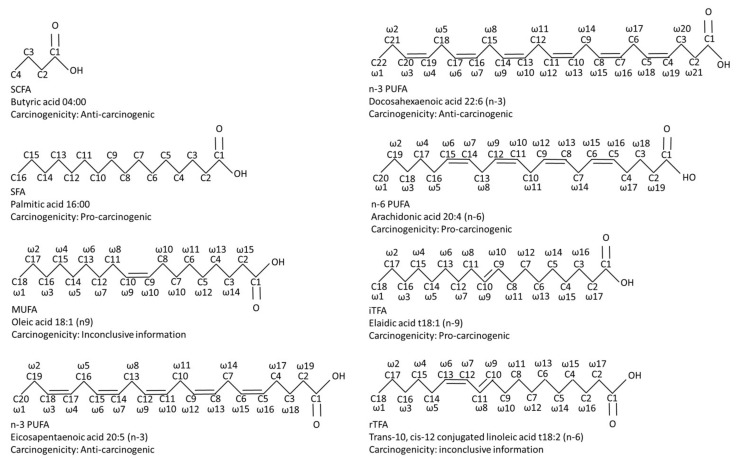
A comparative diagram of the chemical structures of the most investigated fatty acids, including their pro-tumor and anti-carcinogenic properties. Abbreviations: iTFA, industrial trans-fatty acid; MUFA, mono-unsaturated fatty acid; *n*-3 PUFA, omega-3 poly-unsaturated fatty acid; *n*-6 PUFA, omega-6 poly-unsaturated fatty acid; rTFA, ruminant trans-fatty acid; SCFA, short-chain fatty acid; SFA, saturated fatty acid.

**Figure 2 biomedicines-11-00280-f002:**
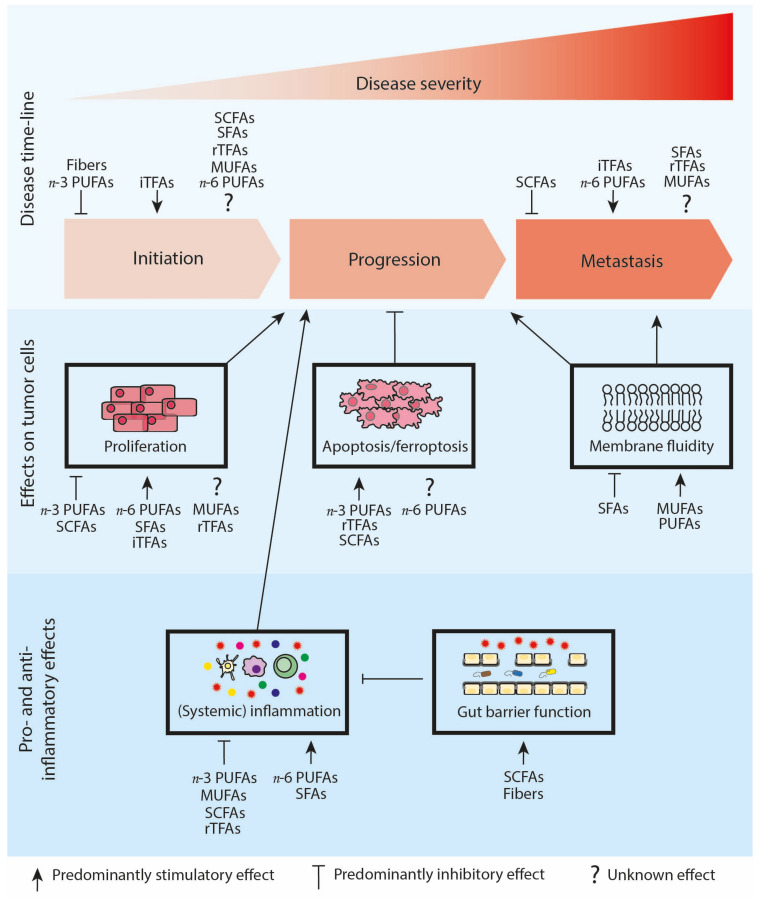
Schematic overview of the relationship between the different fatty acid classes and cancer initiation, progression and metastasis. Cancer risk: iTFAs increase the risk of cancer initiation, while dietary fibers and *n*-3 PUFAs in general seem to decrease the risk of cancer initiation. For SCFAs, SFAs, rTFAs, MUFAs and *n*-6 PUFAs information remains controversial. Cancer progression: *n*-3 PUFAs and SCFAs seem to reduce cancer cell proliferation and stimulate apoptosis/ferroptosis. Opposite, *n*-6 PUFAs, SFAs and iTFAs predominantly show a stimulatory effect on cancer cell proliferation. The effects of MUFAs and rTFAs on cancer cell proliferation are ambiguous, while rTFAs induce apoptosis. Data for *n*-6 PUFAs on apoptosis and ferroptosis are contradictory. Alterations in membrane fluidity influence tumor progression and metastasis. SFAs decrease membrane fluidity, whereas MUFAs and PUFAs stimulate membrane fluidity. SCFAs increase gut barrier functioning, decrease systemic inflammation and, by extension, might reduce tumor progression. MUFAs, *n*-3 PUFAs and rTFAs also reduce systemic inflammation, while SFAs and *n*-6 PUFAs increase systemic inflammation. Cancer metastasis: SCFAs predominantly reduce metastasis, while iTFAs, SFAs and *n*-6 PUFAs seem to stimulate metastasis. The role of rTFAs and MUFAs in metastasis development is not yet understood. Abbreviations: iTFAs, industrial trans-fatty acids; MUFAs, mono-unsaturated fatty acids; *n*-3 and *n*-6 PUFAs, omega-3 and omega-6 polyunsaturated fatty acids; rTFAs, ruminant trans-fatty acids; SCFAs, short-chain fatty acids; SFA, saturated fatty acids.

**Table 1 biomedicines-11-00280-t001:** (Pre)Clinical evidence on the effects of specific fatty acids in the context of cancer progression and metastasis, as well as inflammation.

Type of Fatty Acid	Common Name	Lipid Name	Chemical Name	Preclinical Evidence of Effects on Tumor Cells	(Pre)Clinical Evidence of Pro- and Anti-Inflammatory Effects
SFA	Myristic acid	14:00	Tetradecanoic acid	Post-translational protein myristoylation [37]	
	Palmitic acid	16:00	Hexadecanoic acid	Post-translational protein palmitoylation [38] Promote cancer incidence and proliferation [39] Stimulate metastasis [40] Epigenetic regulation [40]	Trigger pro-inflammatory macrophage activation [41] Trigger TLR4 pro-inflammatory signaling [42,43,44] Trigger NF-κB pro-inflammatory signaling [39,45] Promote pro-inflammatory signaling [39]
MUFA	Palmitoleic acid	16:1 (*n*-9)	Palmitoleate		Suppress pro-inflammatory macrophage activation [46] Reduce HIF-1α expression [46]
	Oleic acid	18:1 (*n*-9)	Octadecenoic acid	Promote cancer cell proliferation [47,48] Reduce cancer cell proliferation [49,50,51] Stimulate cancer cell migration [47] Reduce metastasis [51] Increase angiogenesis [48]	Reduce pro-inflammatory signaling [52,53,54,55]
n3 PUFA	Eicosapentaenoic acid (EPA)	20:5 (*n*-3)	all-cis-5,8,11,14,17 eicosapentaenoic acid	Induce excessive oxidative stress [56]	Reduce pro-inflammatory signaling [57,58,59,60,61]
	Docosahexaenoic acid (DHA, Cervonic acid)	22:6 (*n*-3)	all-cis-4,7,10,13,16,19-docosahexaenoic acid	Induce excessive oxidative stress [62] Induce cancer cell ferroptosis [62] Induce apoptosis via epigenetic regulation [63]	Reduce pro-inflammatory signaling [57,58,59,60,61]
n6 PUFA	Linoleic acid (LA)	18:2 (*n*-6)	all-cis-9,12-octadecadienoic acid	Increase cancer cell proliferation [64] Reduce cancer cell apoptosis [65] Induce cancer cell apoptosis [66,67,68] Induce epigenetic alterations [69]	
	Arachidonic acid (ARA)	20:4 (*n*-6)	all-cis-5,8,11,14-eicosatetraenoic acid	Increase cancer cell proliferation [64] Reduce cancer cell apoptosis [65] Stimulate metastasis [64]	Stimulate inflammation [70,71,72] Induce an immuno-suppressive tumor micro-environment [72]
iTFA	Elaidic acid	t18:1 (*n*-9)	Octadecenoic acid	Decrease survival time [73,74] Stimulate cancer cell proliferation [75] Stimulate invasion [75] Stimulate metastasis [75,76,77]	
rTFA	Trans-palmitoleic acid	t16:1 (*n*-7)	9-Hexadecenoic acid		Reduce pro-inflammatory signaling [78]
	Trans-vaccinic acid	t18:1 (*n*-11)	trans-11-Octadecenoic acid		Reduce pro-inflammatory signaling [78]
	Trans-10, cis-12-conjugated linoleic acid (trans,cis CLA)	t18:2 (*n*-6)	10E,12Z-octadeca-10,12-dienoic acid	Increase cancer cell apoptosis [79] Suppress cancer cell proliferation [79] Increase cancer cell proliferation [80,81,82] Induce anti-metastatic properties [83,84,85] Stimulate metastasis [81]	Reduce pro-inflammatory signaling [86,87]
	cis9, trans11-conjugated linoleic acid (cis,trans CLA)	t18:2 (*n*-6)	9E,11Z-octadeca-9,11-dienoic acid	Increase apoptosis [79] Suppress proliferation [79] Increase proliferation [80] Induce anti-metastatic properties [83,84,85]	Reduce pro-inflammatory signaling [86,87]
SCFA	Acetic acid	02:00	Ethanoic acid	Histone deacetylase inhibition [88]	Reduce pro-inflammatory signaling [89,90]
	Propionic acid	03:00	propanoic acid	Histone deacetylase inhibition [88]	Suppress systemic inflammation via microbiota changes [88] Reduce pro-inflammatory signaling [91]
	Butyric acid	04:00	Butanoic acid	Histone deacetylase inhibition [88,92,93] Suppress cancer cell proliferation [94,95] Trigger apoptosis of cancer cells [93,96,97] Induce anti-metastatic properties [98] Induce epigenetic alterations [63]	Improve intestinal barrier function [99,100,101] Trigger anti-inflammatory immune cells [89,102] Suppress pro-inflammatory immune cells [89,102,103]

Abbreviations: CLA, conjugated linoleic acid, iTFAs, HIF-1α, hypoxia-inducible factor-1α, industrial trans-fatty acids, MUFAs, mono-unsaturated fatty acids; NF-κB, nuclear factor κB, *n*-3 and *n*-6 PUFAs, omega-3 and omega-6 polyunsaturated fatty acids, rTFAs, ruminant trans-fatty acids; SCFAs, short-chain fatty acids; SFA, saturated fatty acids, TLR4 toll like receptor 4 (TLR4).

## Data Availability

Not applicable.

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
