# Peer review of "The Modulatory Effects of Fatty Acids on Cancer Progression"

_biomedicines, 2023, doi:10.3390/biomedicines11020280_

Round 1
Reviewer 1 Report
Westheim et all address the key aspect of tumor heterogeneity influenced by dietary components such as fatty acids on cancer progression. This manuscript has timely relevance in terms of better understanding on the role of different types of fatty acids as pro-tumor or anti-tumor components. This manuscript addresses the need of nutritional intervention by the selective uses of fatty acids against specific cancer types. However, this review is well-written and supported by suitable diagrams.
But technical comments will help to enhance the impact of this review paper and suggestions are listed below.
1. In the introduction, I would suggest to include a paragraph on tumor heterogeneity and various possibilities and components that can modulate tumor microenvironment.
2. A comparative diagram with chemical structure on the different types of fatty acids including pro-tumor and anti-tumor in nature.
3. In the section, “SFAs stimulate tumor progression” authors are encouraged to make distinction from one cancer types to other cancer types such as breast tumor etc.
4. In recent, implications of SFAs and SCFA are suggested in view of epigenetic alterations. Hence, if possible please include a section of FFAs and epigenetic modulation.
5. A though can be discussed also the interplay between FFAs and tumor progression with additional role of microbiotas.
6. In future, the scope of SCFAs and mimetic of SCFAs could be explored. If preclinical and clinical data available on SCFAs and mimetic of SCFAs, then this will make review even more enriched.
Reviewer 2 Report
Figure 1 is not clear. What do mean different types of arrows?
p. 109. "A recent review suggested that the food source rich in SFAs can actually be responsible for the observed differences, with SFAs from (red) meat and palm oil being evidently pro-carcinogenic (35)" - this review is devoted to role of fatty acids in tumor progression and it refers to the another review on the role of fatty acids in tumor progression, it looks strange
Overall, the review contains data on the correlation between dietary fatty acid intake and cancer progression, as well as on the role of fatty acids at the cellular level, but not enough data on their metabolism. Correlation does not always mean a direct relationship between events. It is necessary to present data on fatty acid metabolism in the body, on their bioavailability. The current impression is that fatty acids consumed in the diet enter the body cells in the unchanged form and affect their functions.
